# UBSOFT: A Simulation Platform for Robotic Skill Learning in Unbounded Soft Environments

**Chunru Lin**[1][*], **Jugang Fan**[1][*], **Yian Wang**[1], **Zeyuan Yang**[1], **Zhehuan Chen**[1],
**Lixing Fang**[1], **Tsun-Hsuan Wang**[2], **Xian Zhou**[3], **Chuang Gan**[1,4]
[1]University of Massachusetts Amherst    [2]Massachusetts Institute of Technology
[3]Carnegie Mellon University    [4]MIT-IBM Watson AI Lab

**Abstract:** It is desired to equip robots with the capability of interacting with various soft materials as they are ubiquitous in the real world. While physics simulations are one of the predominant methods for data collection and robot training, simulating soft materials presents considerable challenges. Specifically, it is significantly more costly than simulating rigid objects in terms of simulation speed and storage requirements. These limitations typically restrict the scope of studies on soft materials to small and bounded areas, thereby hindering the learning of skills in broader spaces. To address this issue, we introduce UBSOFT, a new simulation platform designed to support unbounded soft environments for robot skill acquisition. Our platform utilizes spatially adaptive resolution scales, where simulation resolution dynamically adjusts based on proximity to active robotic agents. Our framework markedly reduces the demand for extensive storage space and computation costs required for large-scale scenarios involving soft materials. We also establish a set of benchmark tasks in our platform, including both locomotion and manipulation tasks, and conduct experiments to evaluate the efficacy of various reinforcement learning algorithms and trajectory optimization techniques, both gradient-based and sampling-based. Preliminary results indicate that sampling-based trajectory optimization generally achieves better results for obtaining one trajectory to solve the task. Additionally, we conduct experiments in real-world environments to demonstrate that advancements made in our UBSOFT simulator could translate to improved robot interactions with large-scale soft material. More videos can be found at https://vis-www.cs.umass.edu/ubsoft/.

**Keywords:** Soft-Body Manipulation, Locomotion, Physics Simulation

## 1 Introduction

Soft materials, such as sand and snow, are prevalent in daily life. Although humans routinely interact with these materials on a large scale—from building a snowman to creating art on a sand beach—equipping robots with similar capabilities presents significant challenges. Unlike rigid objects and terrains that have been extensively studied in robotics and Embodied AI research, soft materials like sand and snow possess unique complexities. Their deformable nature leads to intractably high degrees of freedom, making them difficult to perceive and represent accurately. Moreover, their behavior may change depending on the scale at which they are observed, adding an additional layer of complexity. Moreover, its extremely high dimensionality also poses a challenge to the complexity of computational methods including simulation and learning to interact with such materials.

One prevalent method for teaching robots to manipulate and interact with these materials involves collecting extensive data to train the robots in virtual environments with physics solvers dedicated to simulating their physical behaviors [1, 2, 3, 4]. These models are then adapted for real-world applications. However, prior research was often restricted to narrowly defined tasks with confined

---

[*]denotes equal contribution

8th Conference on Robot Learning (CoRL 2024), Munich, Germany.

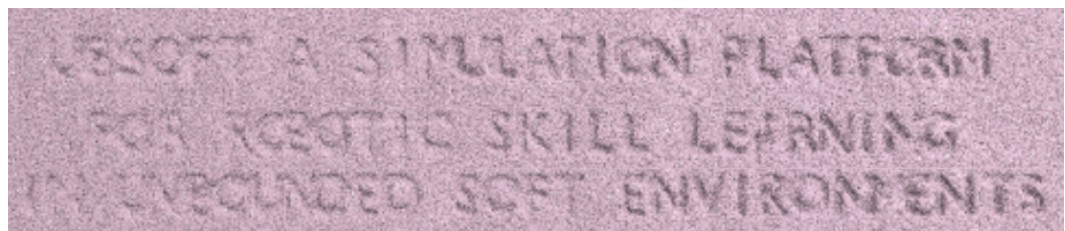

Figure 1: The title of the paper is written on a large scene covered with soft material by a robotic manipulator.

workspace, limiting the task scope to small or simplistic manipulation challenges. For example, Huang et al. [3] focus on small tasks like block lifting or dough reshaping, Xian et al. [4] address fluid manipulation tasks confined in a container, Wang et al. [2] manipulate small pieces of cloth on the table. Yet, these scenarios rarely scale to the complexities encountered in real-world environments; this requires either conducting multiple smaller-scale simulations in an ad hoc manner and meticulously combining them together or, a more intelligent solution: a large-scale yet efficient simulation. For instance, accurately simulating a humanoid robot walking on a sand dune requires simultaneously modeling billions of sand particles, posing significant challenges in terms of both storage capacity and computational speed.

To address these challenges, several recent studies have attempted to simulate large-scale soft materials. A group of work aims to simulate extensive sand environments by using height field [5, 6]. However, this simplification can yield unrealistic results, especially for softer materials like snow which cannot be accurately modeled using 2D height field, or when interaction extends beneath the terrain surface (e.g., deep excavation) is involved. Other research efforts [7] try to mitigate the impact on computational resources by simply freezing particles that are far from agents. Nevertheless, this approach overlooks the fact that material distant from the agent can still move, albeit slowly, and still gradually affect the behaviors of both the agent and the environment. Additionally, this strategy does not fully resolve the issues of storage capacity required for large-scale simulations. Yue et al. [8] developed a more efficient method by adaptively coupling discrete and continuum simulations of granular materials. Despite these advancements, simulating even just within the continuum space remains a costly endeavor. As a result, there still remains a substantial gap in the current literature concerning the simulation of large scenarios filled with soft materials.

To this end, we present UBSOFT, a simulation platform aiming to efficiently simulate large-scale scenes filled with soft materials, enabling real-time and long-horizon robot learning in *unbounded* environments. UBSOFT is a fully differentiable physics simulation platform for robotic skill learning, and supports simulation of giant scenes such as an agent writing the paper title on the sand in Fig. 1 (left) and a robot dog walking on a desert in Fig. 1 (right). We use Material Point Method (MPM) [9], a hybrid particle-grid method to simulate soft materials, and propose a spatially adaptive simulation scheme where simulation resolution dynamically adjusts based on their proximity to the active robotic agent. This enables both efficiency in simulation speed and memory cost, while still leading to accurate simulation behavior of the whole environment. Specifically, we implement a hierarchical grid system and a corresponding particle adjustment mechanism: finer grids and smaller particles are utilized in the region closer to the robot to increase detail and accuracy, while larger grids and particles are used further away to conserve computational resources. In addition, we also present a suite of long-horizon tasks covering both locomotion tasks and manipulation tasks that demonstrate the capability of our platform. Our simulation platform is implemented to be fully differentiable and grants access to gradient information of the system, which has been proven useful for optimizing robot policies when interacting with soft materials in various literature [3, 4, 10]. We conducted evaluations using gradient-based and sampling-based trajectory optimization methods, as well as state-of-the-art reinforcement learning algorithms. The findings indicate that sampling-based trajectory optimization generally delivers the best performance in our context. Reinforcement learning algorithms, which map states to actions, struggle with partial and high-dimensional state representations. Meanwhile, gradient-based methods are less effective on their own due to the highly non-convex landscape and inaccuracies in long-term back-propagation. We summarize our contributions as follows:

- We introduce UBSOFT, a simulation platform designed to facilitate robot learning in spatially extensive environments populated with diverse soft materials. This platform addresses the critical need for comprehensive training in both manipulation and locomotion tasks in expansive environments.

- The core of UBSOFT is a spatially adaptive physics engine that efficiently models various soft materials in large and potentially unbounded environments. We enhance this capability by integrating interactions with rigid materials and incorporating the differentiability of our simulation engine, further supporting the training of robotic agents.

- We present a suite of tasks to benchmark reinforcement learning algorithms and optimization-based methods to comprehensively analyze the performance and challenges of current methods.

## 2   Related Works

**Physics Simulation for Soft Environment.** Recent scholarly interest increasingly focuses on the integration of differentiable simulation and machine learning due to its ability to provide specific environmental feedback through physics. A prevalent approach for building differentiable simulations involves learning forward dynamic models via neural networks [11, 12, 13, 14, 15, 16, 17], which allows for direct gradient propagation through the networks. Although this method typically simulates faster than real-time, it faces challenges such as data dependency, sim-to-real gap, and limitations in handling out-of-distribution scenarios [18]. Another widely adopted method involves implementing physical simulations in a differentiable manner, as seen in [19, 20, 3, 4, 21, 22, 23, 24, 25]. Here, differentiability is achieved using automatic differentiation tools like Taichi [19] and Nvidia Warp [26]. In studies such as Du et al. [27], Li et al. [28], Wang et al. [2], gradients are derived by explicitly computing analytic gradients and back-propagating through the simulation process. These gradient data have been shown to be valuable for tasks that involve the control and manipulation of both rigid and soft bodies [21, 29, 30, 31, 32, 33].

**Interaction with Soft Material.** Robotic manipulation with soft materials has been greatly investigated recently, including manipulation with co-dimensional materials [34, 35, 36, 37, 38, 2, 39, 40], elastoplastic materials [1, 3, 33, 30, 24, 41, 42, 43], granular materials [44, 45], and fluid [4, 46, 47, 48]. Due to the lack of ability to simulate large scenarios, all those works can only deal with tasks in a small scope. There are also groups of works interacting with soft terrains, Wang et al. [32] simulate soft terrain, however, it's still confined within a small scope. Alvarado et al. [49] model soft terrain with only stress-strain model to generate realistic footprints but doesn't deliver real physics. Zhu et al. [5] simulate sand with a height field to achieve sand-vehicle interactions, but it's only for shallow sands and doesn't solve the scaling problem. As a result, we aim to build a simulation platform that can effectively simulate large scenarios in all three dimensions.

## 3   Spatially Adaptive Physics Engine

We develop our physics engine using Taichi [19, 50], a domain-specific programming language embedded in Python for high-performance parallel programming with efficient auto-diff tools. The engine supports various materials, including solid materials (rigid, plastic), soft materials (sand, snow, elastoplastic), and the interactions among them. To enable large-scale and even unbounded soft materials simulation, we design and implement a novel spatially adaptive scheme on top of the Material Point Method, which draws inspiration from the octree structure and is capable of adjusting the simulation precision for different spatial regions based on the robot agent's location.

**Differentiability and Rendering** Following Huang et al. [3], our simulation engine is implemented based on Taichi's autodiff system to compute gradients for simple operations and also implemented an analytical gradient for svd operation [51]. The gradient checkpointing we implemented supports gradient flow over long temporal horizons. UBSOFT also comes with a real-time OpenGL-based renderer and a path tracing renderer implemented using Luisa compute [52].

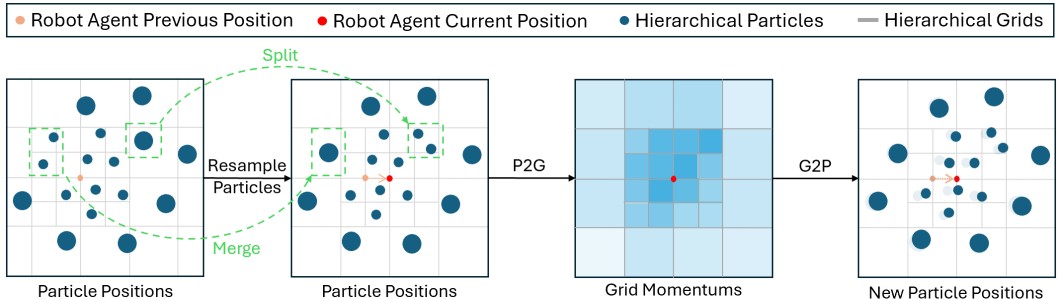

Figure 2: **MPM Simulation with Spatially Adaptive Scheme.** Hierarchical grids are centered around the robot agent and become sparser further out. As the robot moves, these grids move in sync, and the particles are resampled to match the grid cell sizes by splitting large particles in small grids and merging small particles in large grids. Subsequent phases of P2G and G2P adhere to standard MPM protocols but are adapted to accommodate hierarchical grids and particles.

## 3.1 Material Modeling

**Soft body.** We represent soft materials with both Lagrangian particles and Eulerian background grids, using the Moving Least Squares Material Point Method (MLS-MPM) [9]. Specifically, the material properties such as position, velocity, density, volume, and deformation gradients are stored in the Lagrangian particles while the contact with the rigid body and the particle interactions are handled in Eulerian background grids. While our simulation could handle any soft materials that can be simulated by MLS-MPM, we highlight some examples used in our experiments. For instance, **sand** is simulated using a Drucker-Prager plasticity model, which accounts for the material's yield stress and flow rules, effectively capturing the granular flow behavior [53]. Similarly, **snow** is modeled using a combination of elasticity and plasticity to represent its unique compaction and melting characteristics under different conditions [54].

**Rigid body.** In our simulation engine, we employ two types of geometric representations to facilitate the inclusion of rigid bodies, each capable of bidirectional coupling with soft variable structures. The first type, termed "Manipulator", uses Signed Distance Fields (SDFs) to model individual rigid body parts like end-effectors. This method supports differentiable simulation, making it suitable for manipulation tasks. The second type, known as "Articulator", is designed to handle complex articulated rigid bodies featuring multiple links and joints, such as Quadruped robots.

## 3.2 Spatially Adaptive Scheme

We propose a spatial adaptation scheme designed to enhance the efficiency of our simulations. Our rationale is based on the observation that only the immediate vicinity of the robot agent requires high-fidelity simulation, as this area constitutes a small portion of the overall expansive environment. For the vast majority of the environment, which lies distant from the robot, a general overview of movement trends along with rough records resulted from past interaction suffices, eliminating the need for granular detail. Consequently, our strategy focuses primarily on the local environmental information around the robot agent while retaining global environmental information about the footprint from past robot's interaction. Centered on the robot agent, our approach involves employing higher-resolution simulations for proximate areas and lower-resolution simulations for regions that are further away and changing the resolution accordingly as the movement of the robot agent.

**Hierarchical grids.** The granularity of Material Point Method (MPM) simulations depends on the fineness of spatial discretization; finer granularity, achieved through smaller grid cell sizes, results in higher simulation accuracy. Therefore, we introduce a hierarchical grid system as illustrated in Fig.2. Centering on the robot's position, grids of varying granularity are nested, enabling spatial discretization to varying degrees based on the robot's position. Unlike traditional MPM, which applies a uniform granularity across the entire spatial domain, our hierarchical grid *adapts* dynamically according to the distance to the agent. Specifically, with the smallest grid's side length $l_0$, we'll define

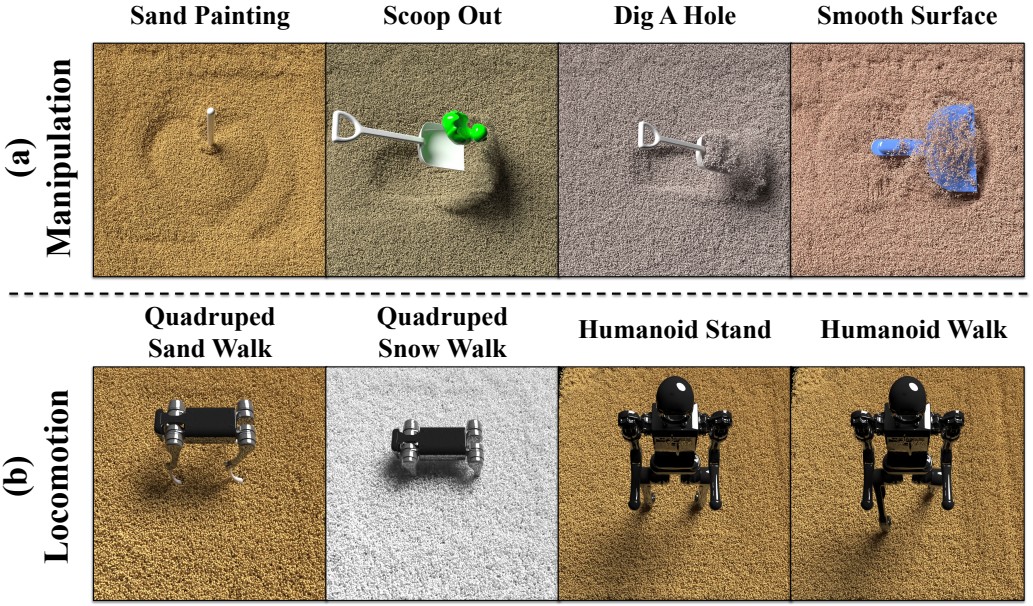

Figure 3: 4 manipulation tasks and 4 locomotion tasks proposed in UBSOFT.

a series of grids with the side length $l_i = 2^i l_0$. Also, we set the number of the grid on each dimension to $2 * k$; in this way, assume the agent centering at $(x, y, z)$, the smallest grids will fill in the area from $(x - kl_0, y - kl_0, z - kl_0)$ to $(x + kl_0, y + kl_0, z + kl_0)$ and subsequently the grids with scale $l_i$ would fill in $(x - kl_i, y - kl_i, z - kl_i)$ to $(x + kl_i, y + kl_i, z + kl_i)$; meanwhile the inner square is still occupied by smaller grids.

**Hierarchical particles.** To maintain an average number of particles per grid cell and also to further save storage space when storing particles, we also adapt the particle size following the grid size. This requires a **particle split** and a **particle merge** process. **Particle split:** Due to the simultaneous movement of particles and the grid, when large particles appear within small grid cells, they will be split into several smaller particles matching the grid cell size to ensure simulation accuracy. The positions of these smaller particles are resampled based on the position of the large particle, while conserving the momentum before and after the splitting, as shown in Fig. 2 *Split*. **Particle merge:** When a certain number of small particles are present within a large grid cell, they are merged into a larger particle to save computational resources. The position of this larger particle is determined by the centroid of the small particles and other maintained information on the particles is averaged according to specific rules, as shown in Fig. 2 *Merge*. Readers are recommended to refer to the code and Appendix A for detailed implementation.

## 4 Benchmark Tasks

### 4.1 Manipulation Tasks

UBSOFT contains 4 diverse manipulation tasks as presented in Fig. 3 where an agent interacts realistically with specific soft material using a tool. Details of each task are described as follows.

**Sand Painting.** This task requires the final surface of the sand to display a heart shape. Therefore, the agent needs to move the tool stirrer, to stir the sand and achieve the specified goal pattern.

**Scoop Out.** A duck partially buried in the sand needs to be scooped out by the agent using a shovel. This task is particularly challenging since the agent must carefully plan the movement path. Otherwise, it is easy to push the duck deeper into the sand.

**Dig A Hole.** The agent needs to mimic a human-like manipulation of the shovel to complete a parabolic-like trajectory, thereby digging a hole in the flat sand surface.

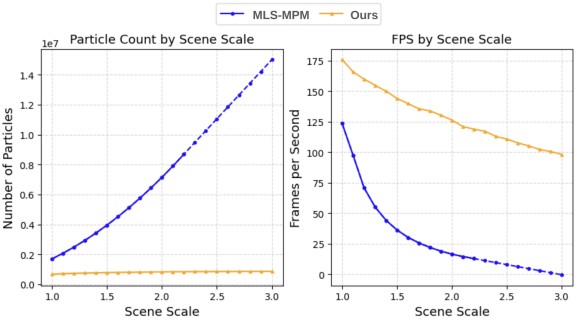

Figure 4: With larger scenes, the spatially adaptive scheme significantly reduces the storage space required and maintains a higher simulation speed. The dotted lines are extrapolated results where transitional MPM fails to simulate.

| Simulation | # of particles ↓ | time(s) ↓ | FPS ↑ | Chamfer distance ↓ |
|---|---|---|---|---|
| **MLS-MPM 256** | **11,612,880** | **579.9** | **8.6** | **-** |
| MLS-MPM 128 | 1,444,000 | 37.1 | 134.8 | 12.97 |
| **Ours** | **712,250** | **27.4** | **182.5** | **5.32** |
| MLS-MPM 64 | 156,620 | 8.2 | 609.7 | 14.15 |
| MLS-MPM 32 | 16,810 | 6.6 | 769.2 | 15.58 |

Table 1: Comparing our spatially adaptive simulation with MLS-MPM of different resolutions.

**Smooth Surface.** Given a sand heap with an overall flat surface but a distinct protrusion, the agent must accurately identify the raised area and move the scraper to approach it. Then, through gentle pressing or other smoothing actions, the agent can finally smooth the surface of the sand.

## 4.2 Locomotion Tasks

**Humanoid Stand.** This task requires a humanoid robot to maintain standing on a sand landscape. The robot must continuously adjust its balance to prevent falling, assessing its ability to adapt to the sandy terrain and maintain stability over an extended period.

**Quadruped (Humanoid) Walk.** A quadruped (humanoid) robot is required to continuously walk on an unbounded sandy pr snowy landscape, resembling a desert. The primary goal is to ensure the robot keeps moving forward while maintaining balance and avoiding getting stuck. To achieve this, the robot must constantly adjust its gait and posture to adapt to the sand's or snow's resistance.

## 5 Experiments

We conduct extensive experiments to answer the following questions:
- How is our proposed simulator compared with existing simulators for soft materials?
- How do different learning algorithms and optimization-based methods perform on our UBSoft benchmark?
- Can policy acquired from the UBSoft Simulator transfer to the real world?

## 5.1 Comparisons with State-of-the-art Simulators

Existing simulators for soft materials mostly rely on MLS-MPM or MPM for simulation. MLS-MPM represents an advancement over MPM by introducing a novel stress divergence discretization, allowing it to run twice as fast as MPM. For a detailed comparison of the speed benchmarking between MPM and MLS-MPM, we recommend readers refer to Section 6.1 of the MLS-MPM paper[9]. Given its faster and more stable performance, we have chosen MLS-MPM as our baseline simulation method.

To assess the **efficiency** of our simulator, we conducted the first experiment to simulate scenes of different scales with MLS-MPM and our spatially adaptive simulation respectively. We monitored the number of particles throughout the simulation process to track the increase in maximum storage required as the scene's spatial scale expanded. Additionally, we benchmarked the speed of our

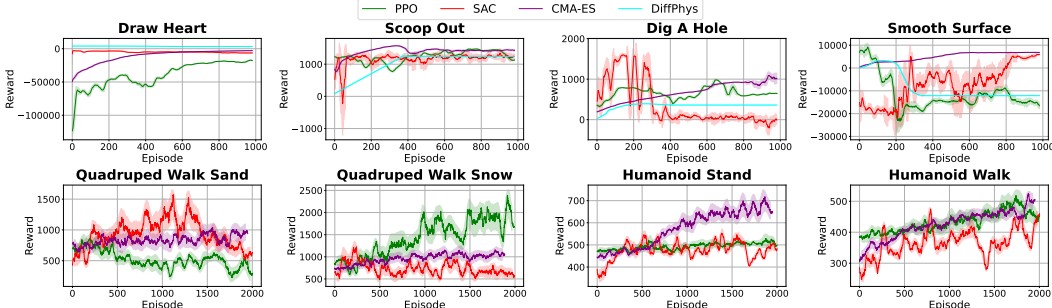

Figure 5: Reward curves for all methods including PPO, SAC, CMA-ES, and Differentiable Physics.

| Task | Sand Painting | Scoop Out | Dig A Hole | Smooth Surface |
|---|---|---|---|---|
| Heuristic | - | 783.4 | 208.1 | 2038.8 |
| PPO | -17280.3 ± 415.4 | 1466.7 ± 22.6 | 980.1 ± 11.2 | **7865.2 ± 725.6** |
| SAC | -3048.2 ± 93.9 | 1265.9 ± 161.0 | **1595.6 ± 48.7** | 7610.4 ± 998.7 |
| CMA-ES | -2860.3 ± 45.7 | **1573.2 ± 5.7** | 1067.0 ± 99.0 | 6844.8 ± 40.7 |
| DiffPhys | **4000.0 ± 37.8** | 1251.0 ± 65.7 | 393.1 ± 0.4 | 3269.0 ± 59.6 |
| | | | | |
| Task | Quadruped Sand Walk | Quadruped Snow Walk | Humanoid Stand | Humanoid Walk |
| PPO | 862.8±94.8 | **2387.4±336.7** | 528.3±32.1 | 518.8±32.7 |
| SAC | **1572.8±245.5** | 1225.0±201.8 | 553.1±34.7 | 475.3±34.8 |
| CMA-ES | 1037.9±123.3 | 1160.6±95.9 | **717.7±47.7** | **524.5±19.3** |

Table 2: The final accumulated reward and the standard deviation for each method.

simulator against MLS-MPM. As in Fig. 4, our method significantly reduces the storage space required for larger scales and achieves a tenfold speedup over MLS-MPM as the scene scale increases.

To further evaluate the **realism** of our simulator, we conducted a comparative analysis between our spatially adaptive simulator and MLS-MPM of different resolutions (256, 128, 64, 32). The simulated scene is a rigid cube of size $0.1 \times 0.1 \times 0.1$ falling into a $1 \times 1 \times 0.2$ sand and the simulation runs 5000 timesteps. Although simulation realism is challenging to quantify precisely, in the field of simulation, finer granularity yields higher accuracy, bringing results closer to reality. Therefore, we used the MLS-MPM 256 simulation as the reference and calculated the Chamfer distance between the particle sets to quantitatively assess the realism of our simulation. The results are presented in Table 1 ranked by time and space consumption. UBSOFT achieved a reduction in time and space consumption by over an order of magnitude. Furthermore, our simulation closely matches the performance of MLS-MPM 256, while requiring only half the space and less time than MLS-MPM 128.

## 5.2 Benchmark different policy learning methods

Our goal is to establish a foundation for skill learning in large-scale environments filled with soft materials. Leveraging our simulation platform, we introduce several tasks tailored to this context and benchmark existing algorithms to broadly assess their effectiveness in these tasks. These experiments will lay the groundwork for future research on solving tasks involving large-scale soft materials.

We benchmark several methods for manipulation and locomotion tasks proposed in section 4. These include a gradient-based trajectory optimization method using differentiable physics (**DiffPhys**), a sampling-based trajectory optimization method known as Covariance Matrix Adaptation Evolution Strategy (**CMA-ES**) [55], model-free Reinforcement Learning (RL) algorithms such as Soft Actor-Critic (**SAC**) [56] and Proximal Policy Optimization (**PPO**) [57], and a heuristic agent manually designed by the authors. For the implementation of these algorithms, we utilize the stable-baselines library [58] for RL algorithms and refer to [59] for the implementation of CMA-ES. Notably, for locomotion tasks, we do not use DiffPhys in the experiments due to its current limitation in handling gradients for articulated objects. Following Wang et al. [2], we compare the reward of the best trajectory of different methods.

**Implementation Details.** For manipulation tasks, we use a 4-layer grid of size $4 \times 64^3$ with resolutions of $256, 128, 64, 32$, the spatial scale is $1 \times 1 \times 0.2$, and the simulation step is set to $0.002$s. We use an 8-layer grid of size $8 \times 32^3$ with resolutions from 128 to 1 and a spatial scale of $10 \times 10 \times 0.5$ for the locomotion tasks.

**Evaluation metrics.** We utilize a human-defined reward function as the direct evaluation metric because it most accurately reflects the training objectives. Rewards are designed based on Chamfer distance for Sand Painting, height variation and scooped particles for Dig A Hole and Scoop Out, and particles above a threshold for Smooth Surface. While for various locomotion challenges, the reward is derived directly from the state of the rigid bodies. The detailed reward function design is elaborated in Appendix B.3.

**Discussions.** Generally, the heuristic agents struggle to finish the task compared to other agents, even with extensive human involvement. RL algorithms generally struggle with manipulation tasks. This difficulty arises primarily because soft materials have infinite degrees of freedom, which are downsampled to create the observation space. Thus, the observation space cannot fully capture the current state, especially when it needs to represent subtle geometries such as **Sand Painting**.

Compared to reinforcement learning algorithms, the sampling-based trajectory optimization algorithm performs generally better. This is because reinforcement learning algorithms learn a policy that maps states to actions, where the state space is very partial and of high dimensionality. In contrast, CMA-ES only learns a trajectory that achieves the best performance starting from the initial state, which can be much easier than learning an entire policy.

We find that directly applying gradient-based optimization using differentiable physics simulation is less effective. The gradient becomes inaccurate due to clipping during long-term back-propagation. However, when it comes to the task **Sand Painting** that requires subtle manipulation skills to achieve and have continuous contacts, the differentiable physics-based trajectory optimization method would outperform others.

### 5.3 Sim-to-Real Transfer

We selected the tasks of sand painting and scooping to establish real-world experiments using an XArm, exploring open-loop transfer. For each task, we first build the digital twins of the real-world scenario in our simulator, by meticulously adjusting the sand model with a parameter of Young's modulus $E = 1e6$, Poisson's ratio $\nu = 0.2$, density $\rho = 1000.0$ and friction angle $\alpha = 45$ to align the observation in the simulation and in the real-world. After that, we optimize the trajectory of the manipulator in our simulated environments and then directly deploy it on the real robotic arm to replicate the trajectory we have in the simulation. We find our trajectory optimized in the UBSoft simulator works well in the real world. As shown in Appendix, the robotic arm successfully writes the letters "CoRL" in the sand, and scoops out the cube from the sand, proving the successful open-loop transfer from UBSoft simulator, which indicates the realistic and potential of our simulator with a small sim-to-real gap. Furthermore, advancements achieved in UBSOFT may also translate into tangible improvements for robots operating in environments dominated by expansive soft materials.

## 6 Conclusion

We introduced UBSOFT, a new simulation platform for benchmarking robot manipulation and locomotion skills learning in large environments filled with diverse soft materials. At the heart of UBSOFT is a spatially adaptive physics engine that efficiently simulates soft materials in extensive, potentially unbounded environments. This engine utilizes an octree structure that dynamically adjusts to the robot's movements. We have further enhanced the platform by enabling interactions with rigid materials and incorporating the differentiability of our simulation to aid in robotic skill learning. The designed manipulation and locomotion tasks rigorously evaluate the performance of existing methods, providing insights for the development of more capable and resilient robots in expansive settings.

**Acknowledgment** This project was supported by NSF IIS-2404386.

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

# A    Implementation Details on the UBSOFTEngine

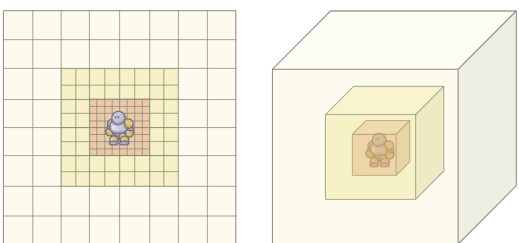

Figure 6: Hierarchical Grids for Spatially Adaptive Scheme.

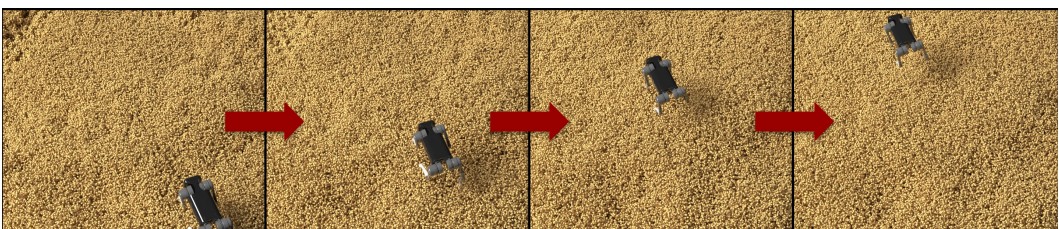

Figure 7: A robot dog walks in a desert with dunes.

## A.1    Spatial Adaptive Scheme

To improve the efficiency of our proposed algorithm, we implement a dynamic data structure based on tagging and amortized restructuring to support the deletion and addition of particles during the resampling process.

The hierarchical grid system introduced in section 3.2 is illustrated in Figure 6. Centering on the robot's position, grids of varying granularity are nested, enabling spatial discretization to varying degrees based on the robot's position. Unlike traditional MPM, which applies a uniform granularity across the entire spatial domain, our hierarchical grid *adapts* dynamically according to the distance to the agent.

With the proposed spatial adaptive scheme, UBSOFT is able to efficiently simulate large-scale scenes filled with soft materials, enabling real-time and long-horizon robot learning in *unbounded* environments, such as a robot dog walking on a desert in Figure 7.

# B    UBSOFT Tasks and Evaluation Details

## B.1    Task Representation

**Task Formulation.** We design two sets of tasks: manipulation tasks and locomotion tasks in our task collection. The two types of task share the same formulation, a finite-horizon Markov Decision Process (MDP), which contains state space $\mathcal{S}$, action space $\mathcal{A}$, reward function $\mathcal{R} : \mathcal{S} \times \mathcal{A} \times \mathcal{S} \to \mathbb{R}$, and transition function $\mathcal{T} : \mathcal{S} \times \mathcal{A} \to \mathcal{S}$. The transition function is determined by the forward simulation. The loss and reward function for each task is discussed in Section 5. The goal of the robot agent is to find a policy $\pi(a|s)$ that produces action sequences to maximize the expected total return $E_\pi[\sum_{t=0}^{T} \gamma^t \mathcal{R}(s_t, a_t)]$ where $\gamma \in (0, 1)$ is the discount factor and $T$ is the horizon of the task.

**State Space and Action Space.** In manipulation tasks, we suppose the robot agent is a robot arm with an end-effector and the state space $\mathcal{S} = \mathcal{S}_A + \mathcal{S}_P$ includes two parts. $\mathcal{S}_A \in \mathbb{R}^{2d}$ represents the pose and velocity information of the end-effector of the robotic arm in the scene, where $d$ is the degree of freedom in the specific task. $\mathcal{S}_P \in \mathbb{R}^{N_P \times 6}$ is the deformation state of non-rigid objects with a

particle-based representation, where $N_P$ is the max possible number of particles in the simulation using our Spatially Adaptive MPM method and 6 represents the 3D position and 3D velocity of the particles. Here we omit the robot arm's link structure because arm movements can be derived from end-effector trajectory by inverse kinematics methods. The action space $\mathcal{A} \in \mathbb{R}^d$ is then defined as the $d$D pose change of the end-effectors. For locomotion tasks, the state $\mathcal{S} = \mathcal{S}_A$ considers only the robot agent since the focus is on the robot's movement.

**Observation.** To facilitate learning and optimization algorithms, we employ stratified sampling to avoid the intractably huge state space. We sample different numbers of particles from the scene according to its grid layer, where more particles from the layers closer to the robot agent are sampled since local states are of more importance for policy learning. We sample $N_S = \sum_{l=1}^{L} N_l = 200$ particles from the scene then stack their positions and velocities to form a $N_S \times 6$ vector as part of the observation, where $L$ is the number of hierarchical grid layers in the scene and $N_1 > N_2 > ... > N_L$. Then we incorporate the robot agent's pose to form the observation of a dimension of $N_S \times 6 + N_J$.

## B.2 Task Details

In RL algorithms, we employ stratified sampling and downsampling from all particles to construct the observation input.

In all manipulation tasks, we utilize a simulation step of 2e-4 seconds, and we establish a maximum action range specific to each task to ensure system stability. In locomotion tasks, the simulation step is 2e-3 seconds composed of 10 sub-steps for easier control policy learning. All tasks run faster than real time (more than 60 FPS where each frame is a simulation step) on a laptop computer equipped with an Nvidia RTX 4090 GPU and an Intel i9 CPU.

## B.3 Reward Design

In each task, the reward is defined as $\mathcal{R} = -\alpha \mathcal{L} + \beta$, where $L$ is the total loss of the entire episode, $\alpha$ and $\beta$ are task-specific constant for reward scaling.

**Sand Painting.** We use the state of particles in the ground-truth policy as the target state, compute the chamfer distance $d_t$ between the current state and the target state at each time step, and sum them up as the total loss, $\mathcal{L} = \Sigma_t d_t$.

**Scoop Out.** This task has two phases, inserting the shovel into the sand and then lifting the shovel to scoop out the duck. We found that simple matching with goal patterns is hard to learn and not robust enough. Therefore, we divided the learning process into two stages as well. In the first stage, the goal is to encourage the shovel to approach a position under the duck, thus $l_1 = \|p_{shovel} - p_t\|$, where $p_t$ is dynamically computed as $p_t = p_{duck} - (0, 0, 0.03)$. In the second stage, the goal is to encourage the shovel to lift and scoop out the duck, thus $l_2 = -z_{shovel} + n_{d,t}$, where $n_{d,t}$ is the number of particles within 0.1m around the duck at time step $i$. Finally, total loss is computed as $\mathcal{L} = \Sigma_t(\sigma_t \cdot l_1 + (1 - \sigma_t) \cdot l_2)$, where $\sigma_t$ indicates whether step $t$ is in the first stage.

**Dig A Hole.** Similar to the Scoop Out task, this task also consists of two stages. In the first stage, the loss function $l_1 = |z_{shovel} - z_p|$ encourages the shovel to descend, where $z_p$ is the z-coordinate of a predefined point below the sand surface. In the second stage, the loss function $l_2 = -(z_{shovel} - z_p)$ encourages the shovel to ascend. The final total loss is computed the same way as in Scoop Out, $\mathcal{L} = \Sigma_t(\sigma_t \cdot l_1 + (1 - \sigma_t) \cdot l_2)$, where $\sigma_t$ indicates whether step $t$ is in the first stage.

**Smooth Surface.** In this task, we use the number of particles above a predefined threshold as the metric. Specifically, the loss is defined as $\mathcal{L} = \Sigma_{t>1}(n_{s,t} - n_{s,t-1})$, where $n_{s,t}$ is the number of particles above the average sand surface height.

**Quadruped Walk.** At each step, we compute the distance $d_t$ between the current state and the target state, represented by the pelvis joint state. The target state is defined as the robot's state that maintains a forward velocity close to 1 m/s. The total loss $\mathcal{L} = \Sigma_t d_t$ is the sum of these distances.

**Humanoid Stand.** To maintain the humanoid robot in a standing pose, we compute the delta height between the current state and the initial state, represented by the pelvis joint state $d_{p,t}$ and the head joint state $d_{h,t}$. The total loss $\mathcal{L} = \Sigma_t(d_{p,t} + d_{h,t})$ is the sum of these delta heights.

**Humanoid Walk.** The target state is defined as the humanoid robot's state that maintains a forward velocity close to 1m/s without falling to the ground. At each step, we compute the distance $d_t$ between the current and the target pelvis state, reflecting the forward velocity. Additionally, we compute the delta head joint height $d_{h,t}$, representing the standing pose. The total loss $\mathcal{L} = \Sigma_t(d_t + d_{h,t})$ is the sum of the forward distances and the delta heights.

## C   Additional Experiments Results

| Task | Quadruped Sand Walk | Humanoid Sand Stand | Humanoid Sand Walk |
|---|---|---|---|
| PPO | 862.8±94.8 | 528.3±32.1 | 518.8±32.7 |
| SAC | **1572.8±245.5** | 553.1±34.7 | 475.3±34.8 |
| CMA-ES | 1037.9±123.3 | **717.7±47.7** | **524.5±19.3** |

| Task | Quadruped Snow Walk | Humanoid Snow Stand | Humanoid Snow Walk |
|---|---|---|---|
| PPO | **2387.4±336.7** | 507.0±12.0 | 460.2±21.4 |
| SAC | 1225.0±201.8 | **753.3±61.3** | 530.9±37.2 |
| CMA-ES | 1160.6±95.9 | 716.6±39.9 | **554.1±30.6** |

| Task | Quadruped Elastic Walk | Humanoid Elastic Stand | Humanoid Elastic Walk |
|---|---|---|---|
| PPO | **2387.4±514.6** | 555.7±36.6 | 485.1±32.2 |
| SAC | 1214.6±276.6 | 653.4±51.2 | 579.7±37.3 |
| CMA-ES | 1224.4±156.5 | **686.2±31.9** | **648.8±40.4** |

Table 3: The final accumulated reward and the standard deviation for each method.

### C.1   Locomotion Task Results on More Materials

To further investigate our environment, we conduct experiments with locomotion tasks on more materials, including sand particles, snow particles, and elastic particles. The results are presented in Table 3 and Figure 8. Similar to the results reported in the main paper, reinforcement learning algorithms generally struggle with these tasks. In particular, PPO generally fails on humanoid standing tasks across all surfaces. On the other hand, the sampling-based trajectory optimization algorithm, represented by CMA-ES, typically achieves better performance. This pattern is further amplified by the large and complex state and action spaces associated with locomotion tasks. Additionally, maintaining a specified pose or moving forward consistently remains challenging for end-to-end training, especially on diverse rough surfaces. Consequently, developing an improved policy for locomotion tasks remains a significant challenge.

### C.2   Sim-to-Real Transfer

As depicted in Fig. 9, the robotic arm successfully writes the letters "CoRL" in the sand, and scoops out the cube from the sand, proving the successful open-loop transfer from UBSoft simulator, which indicates the realistic and potential of our simulator with a small sim-to-real gap.

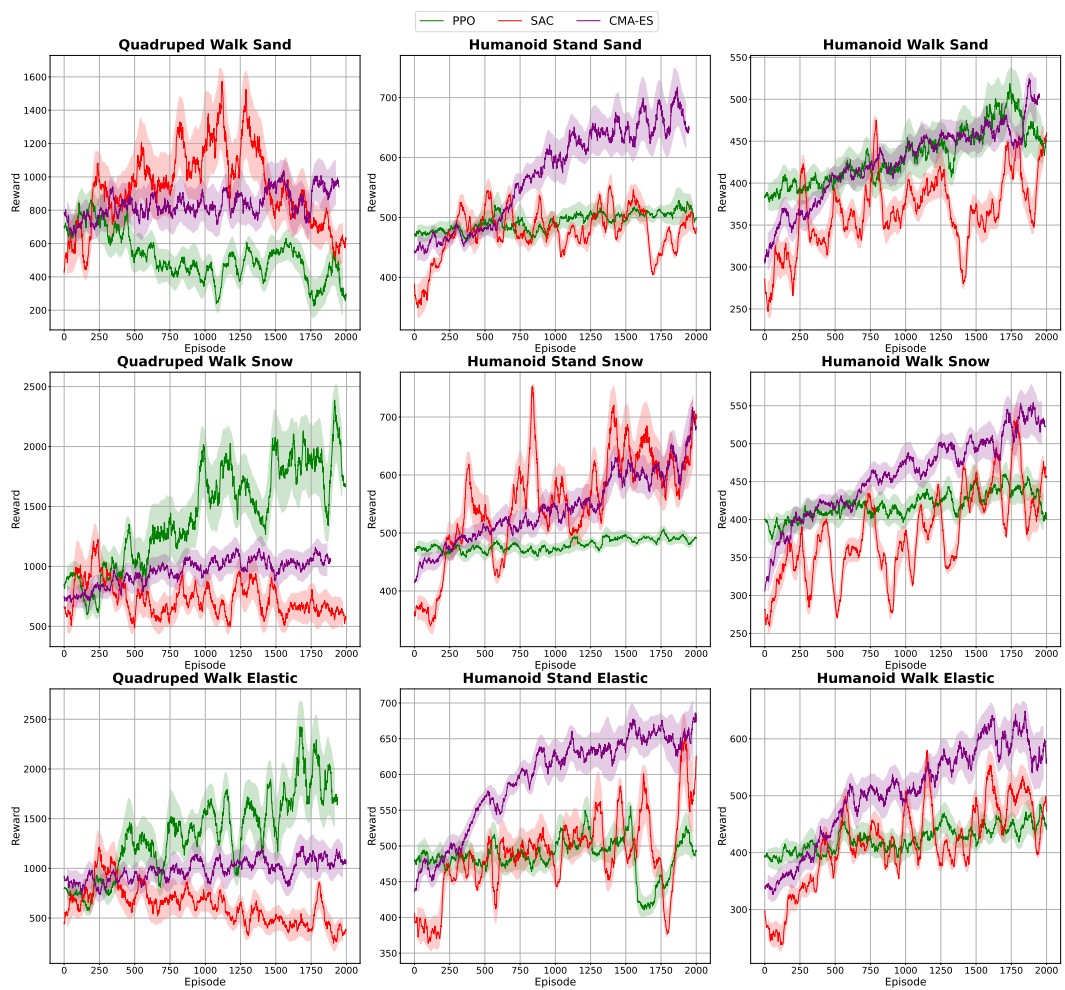

Figure 8: Reward curves for all methods on locomotion tasks, including PPO, SAC, and CMA-ES.

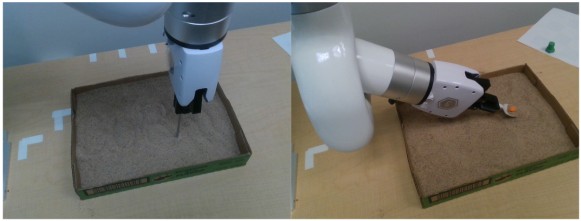

Figure 9: Sand painting and scooping task rollouts in the real world using an XArm.

