# OpenReview forum: "UBSoft: A Simulation Platform for Robotic Skill Learning in Unbounded Soft Environments"
_robot-learning.org/CoRL/2024/Conference — CoRL 2024_

### Official Review · Reviewer_2B5z · 2024-07-16
**Impressive soft material simulator and benchmark**

**Originality:** 3
**Technical Quality:** 4
**Clarity Of Presentation:** 4
**Potential Impact:** 3
**Recommendation:** 3
**Confidence:** 3

**Review:**

The paper has a few noteworthy strengths. First of all, the proposed simulator seems pretty revolutionary and also quite realistic. The authors developed their physics engine using Taichi, and invented a novel spatially adaptive scheme on top of the Material Point Method to support unbounded soft material. The simulator can also support both rigid bodies (e.g. robots) and soft bodies (e.g. sand, snow, etc). Figure 5 is worth highlighting because it very clearly shows the benefits of spatially adaptive scheme in terms of storage and speed.

The paper also proposes a benchmark of 4 manipulation tasks and 4 location tasks, as well as a realistic stratified sampling method to avoid the intractably huge state space. The experimental evaluation is relatively solid, as the authors compare representative approaches for these 8 tasks, gradient- and sampling- based trajectory optimization method, and on- and off-policy RL methods. Real world experiments are also conducted to demonstrate the consistency and realism of the UBSoft simulator.

However, the paper is not without room for improvement. First of all, the tasks showcased are still quite “specialized” per se. For example, can the simulator support many rigid bodies (e.g. 100 different everyday objects) and many soft bodies (e.g. snow, sand, water, mixed together) at the same time? What’s the performance implication? Furthermore, for experimental evaluation, it’s unclear whether 1000 or 2000 episodes are enough for training RL methods, i.e. is performance going to go up if more data is given? Also, more details about the real-world experiments to really demonstrate the realism of UBSoft. To be honest, it’s still unclear how realistic the simulation is, e.g. when looking at the videos on the website (https://ubsoft24.github.io/), it appears that the flying particles move slower than expected in tasks “Scoop Out”, “Dig a Hole” and “Smooth Surface”.

**Quality Of The Limitations Section:**

3

**Questions For Rebuttal:**

- Can the simulator support many rigid bodies? How would that impact run time? E.g. burying 100 rigid bodies inside the sand dunes.
- Why is DiffPhys unable to handle gradients for articulated objects?
- Have you tried to train the RL agents longer, with more episodes? 1 or 2K episodes seem very restrictive.
- Do you have more quantitative results for the real-world experiments? Do you observe any performance degradation? If so, what kind of degradation?

**Robotics Focus:**

4

**Summary Of Paper:**

The paper presents UBSoft, a new simulation platform designed to support unbounded soft environments (sand, snow, etc) for robot skill acquisition. Via spatially adaptive resolution scales, the proposed framework can markedly improve storage and computation efficiency, which enables simulating large-scale scenarios involving soft materials. The authors also establish a set of benchmark tasks, including both locomotion and manipulation tasks, and conduct experiments to evaluate various RL algorithms and trajectory optimization techniques, both gradient-based and sampling based. Finally, the authors conduct real-world experiments to demonstrate the realism of the UBSoft simulator.

**Summary Of Recommendation:**

I would recommend weak accept since the proposed simulator seems to be quite advanced and useful for the wider community that might want to study robot interaction with soft material.

---

### Official Review · Reviewer_P9RJ · 2024-07-23
**Good work - but needs extensive and rigorous evaluation**

**Originality:** 2
**Technical Quality:** 2
**Clarity Of Presentation:** 3
**Potential Impact:** 2
**Recommendation:** 3
**Confidence:** 3

**Review:**

The paper addresses the impactful problem of simulating large-scale soft materials for robotic interactions, proposing a differentiable simulator based on Taichi. Leveraging the existing Material Point Method (MPM), the novelty lies in the spatially adaptive simulation, adjusting the resolution of points based on their proximity to the robot agent. The paper demonstrates learning agents using the proposed simulator.

While the proposal is sound, the experiments do not convincingly demonstrate the advantages of the proposed method compared to existing simulators for deformable materials. Several critical aspects need further clarification and validation.

Strength:
- The problem studied is significant for robotic interactions with soft materials.
- The use of spatially adaptive simulation presents a potentially novel approach.

Weaknesses:
- Lack of comparative analysis with existing simulators for deformable materials.
- Insufficient experimental evidence to highlight the advantages of the proposed method.
- Need for detailed explanation of the distinction between the proposed simulation and Taichi, focusing on the spatial adaptive module.

Recommendations:

- Provide comparative experiments with existing simulators to showcase the advantages of the proposed method.
- Conduct an ablation study to illustrate the significance and impact of the spatially adaptive module, including potential speed-ups.
- Evaluate the realism of the simulation relative to real-world scenarios, addressing any design choices that may hinder realism.
- Expand the related work section to clarify the novelty of the approach, comparing it with state-of-the-art (SOTA) simulators and similar ideas in the literature.
- Use metrics and baselines to convincingly convey the significance of the proposal, comparing the performance of fixed methods using different simulations.

**Quality Of The Limitations Section:**

2

**Questions For Rebuttal:**

I think the paper has a lot of rooms of improvement in terms of experiments:

1. Comparison with Existing Simulators:
Need experiments and metrics to answer, how does the proposed method compare with existing simulators for deformable materials.

2. Distinction and Novelty:
What is the distinction of the proposed simulation from Taichi? Is the spatially adaptive module the primary novelty? If so, can you present experiments illustrating its significance, such as speed-up measurements?

3. Realism and Design Choices:
How do you measure the realism of the simulation compared to the real world? Do any design choices hinder the realism of the simulation?

4. Baseline Comparisons:
- Can you provide an ablation study comparing more agents that do not depend on the proposed simulator (e.g., policies trained using different simulators)?


Also could you clarify:

5. Related Work and Novelty Clarification:
- Can you elaborate more on related work to clarify the novelty of your approach? What problems do existing SOTA simulators face, and how does your method address them? Is there any literature employing similar ideas to the spatially adaptive module, or is it completely new?

6. Experimental Setup and Task Clarification:
- Could you elaborate on why the proposed tasks in Section 4.2 cannot be solved by simple replay or heuristic agents (and include them in the comparison)?

**Robotics Focus:**

4

**Summary Of Paper:**

The paper proposes a differentiable simulator based on Taichi for simulating large-scale soft materials, introducing a spatially adaptive simulation technique that adjusts resolution based on proximity to the robot agent.

**Summary Of Recommendation:**

To strengthen the paper, comparative experiments with existing simulators and a clearer distinction of the novelty are needed. Additionally, evaluating the realism of the simulation, expanding related work, and providing detailed experimental clarifications will better demonstrate the proposed method's advantages.

---

### Official Review · Reviewer_gbiB · 2024-07-24
**The contribution is effective from the qualitative results but lacks convincing sim and real experiments.**

**Originality:** 2
**Technical Quality:** 3
**Clarity Of Presentation:** 4
**Potential Impact:** 3
**Recommendation:** 3
**Confidence:** 3

**Review:**

Strengths:
1. The paper is written well.
2. The reviewer thinks that the problem is important to simulation research and development. The benchmark tasks of locomotion, scoop out tasks are good evaluation tasks and interesting to the reviewer as it has many applications.
3. The formulation of the adaptively scaling the resolution of both the lagrangian particles and Eulerian grids is novel, and proves to be qualitatively effective. The videos in the supplementary and the qualitative results from the paper show how the deformation is constrained to the local areas and that the resolution scaling does not promote deformation movements away from the region of manipulation.

Weakness:
1. The contribution seems to be minor in the reviewer’s eyes — resolution scaling around the robot.
2. The experiments benchmark different algorithms in the simulator. The reviewer would like it if it was addressed why these experiments were done.
3. From the paper, in the real world experiments it is simply mentioned that "similar policies are implemented from those in UBSoft" and deployed in reality. The reviewer assumes this means no sim2real transfer. There is no transfer from sim2real and no comparison between different simulation strategies to show the clear effectiveness of the method. For example, transfer to real world deployments of tasks like the locomotion task would show effectiveness of training a locomotion controller on UBSoft. The reviewer understands this may be hard, and would've liked to see some transfer to the real world from simulation (unless the reviewer has misunderstood the real experiments sections).
4. There’s no comparison on how the simulation performs in comparison to MLS-MPM[2], MPM[1] or other existing simulators for this task.

Nitpick:
On line 65, “scehem” is spelled incorrectly.

[1] "Application of a particle-in-cell method to solid mechanics"
[2] "A Moving Least Squares Material Point Method with Displacement Discontinuity and Two-Way Rigid Body Coupling"

**Quality Of The Limitations Section:**

3

**Questions For Rebuttal:**

Comments:

Additional details on how the physics evolves when the grid sizes/particle sizes change would make the contribution more concrete.

Questions

Does the simulation efficiency section compare MPMSA with MPM or MLS-MPM? If MPM, how does the method fair in comparison to MLS-MPM?

From the real experiments, is this sim2real or new policies are trained on the real robot? What does it mean for “exhibit trends similar to those observed in UBSOFT”? The reviewer thinks this does not have any backing quantitatively.

**Robotics Focus:**

4

**Summary Of Paper:**

The paper addresses the problem of simulating soft materials in unbounded regions, as opposed to previous methods that simulate bounded areas because of computational challenges. The authors propose adapting the resolution scales (of the grids and particles) given the robot’s location to adapt MPM [1] based solutions that use fixed grid and particle sizes. Furthermore, 4 manipulation and 4 locomotion benchmark tasks on sand based environments are presented. Different task algorithms are compared with task reward metrics. A qualitative real world study is also provided.

**Summary Of Recommendation:**

Although the paper has a clear conceptual contribution and show’s effective results to the claims made, the paper lacks experiments with other baseline methods as mentioned in the detailed review. The reviewer also found the technical contribution to be minor. If more experiments are shown in comparison to MLS-MPM and questions/comments are addressed, the reviewer would be more positive about the paper.

---

### Author Rebuttal · Authors · 2024-08-12

We thank all the reviewers and AC for the insightful comments and constructive suggestions to strengthen our work. We uploaded the revised paper and updated experimental result figures in the rebuttal zip file. We will continue to refine our manuscript. Thank you.

---

### Decision · Program_Chairs · 2024-09-04

**Decision:**

Accept

**Comment:**

# Strengths:
Multiple reviewers acknowledge the following strengths in the paper
- the writeup clarity
- the importance of the problem to simulation and robotics research
- novelty around spatially adaptive simulation, adjusting the resolution of points based on their proximity to the robot agent.


# Weakness:
Reviewers have also recognized the following weaknesses
- lack of comparison to MLS-MPM[2], MPM[1] or other existing simulators.
- Clarity around its differences wrt to Taichi.
- experiments do not convincingly demonstrate the advantages of the proposed method compared to existing simulators for deformable materials
- Realism of the simulation
- Ambiguity around methods and results - "similar policies are implemented from those in UBSoft" and deployed in reality. The reviewer assumes this means no sim2real transfer. There is no transfer from sim2real and no comparison between different simulation strategies to show the clear effectiveness of the method."

# Recommendations:
It is recommended to extend the comparisons and carefully strengthen the claims. Reviews have provided excellent recommendations on how to improve. Listing a few below -
- Comparison with Existing Simulators: Need experiments and metrics to answer, how the proposed method compares with existing simulators for deformable materials.
- Additional details on how the physics evolves when the grid sizes/particle sizes change would make the contribution more concrete.
- From the real experiments, is this sim2real, or new policies are trained on the real robot? What does it mean to “exhibit trends similar to those observed in UBSOFT”? The reviewer thinks this does not have any backing quantitatively.
-  What is the distinction of the proposed simulation from Taichi? Is the spatially adaptive module the primary novelty? If so, can you present experiments illustrating its significance, such as speed-up measurements?

# Rebuttal
Authors and reviewers engaged in healthy discussions during rebuttals. Post rebuttals, all reviewers are aligned on the recommendation of the paper. In support of their feedback, AC recommends accepting the paper. Reviewers should address all the pending commitments from the rebuttals and prepare the final manuscript accounting for all the received feedback.